Struo2: efficient metagenome profiling database construction for ever-expanding microbial genome datasets

Youngblut Nicholas D. nyoungblut@tuebingen.mpg.de
Ley Ruth E.
Microbiome Science, Max Planck Institute for Developmental Biology , Tuebingen , Baden Wurttemberg , Germany
Parks Donovan
Electronic publication date: 2021 Sep 16
Publication date: 2021
Volume: 9
Electronic Location ID: e12198
Received 2021 Aug 4; Accepted 2021 Aug 31
Copyright: ©2021 Youngblut and Ley
Copyright year: 2021
Copyright holder: Youngblut and Ley
License: This is an open access article distributed under the terms of the Creative Commons Attribution License, which permits unrestricted use, distribution, reproduction and adaptation in any medium and for any purpose provided that it is properly attributed. For attribution, the original author(s), title, publication source (PeerJ) and either DOI or URL of the article must be cited.
License URL: https://creativecommons.org/licenses/by/4.0/

Keywords: Metagenome, Database, Profiling, GTDB

Funding: This work was supported by the Max Planck Society. The funders had no role in study design, data collection and analysis, decision to publish, or preparation of the manuscript.

==============================
Mapping metagenome reads to reference databases is the standard approach for assessing microbial taxonomic and functional diversity from metagenomic data. However, public reference databases often lack recently generated genomic data such as metagenome-assembled genomes (MAGs), which can limit the sensitivity of read-mapping approaches. We previously developed the Struo pipeline in order to provide a straight-forward method for constructing custom databases; however, the pipeline does not scale well enough to cope with the ever-increasing number of publicly available microbial genomes. Moreover, the pipeline does not allow for efficient database updating as new data are generated. To address these issues, we developed Struo2, which is >3.5 fold faster than Struo at database generation and can also efficiently update existing databases. We also provide custom Kraken2, Bracken, and HUMAnN3 databases that can be easily updated with new genomes and/or individual gene sequences. Efficient database updating, coupled with our pre-generated databases, enables “assembly-enhanced” profiling, which increases database comprehensiveness via inclusion of native genomic content. Inclusion of newly generated genomic content can greatly increase database comprehensiveness, especially for understudied biomes, which will enable more accurate assessments of microbiome diversity.

Introduction

Utilizing shotgun metagenomics for microbiome research is increasing in popularity due to the reduced costs and scalability of sequencing and the improved ease of sequence library generation (Sczyrba et al., 2017; Breitwieser, Lu & Salzberg, 2019). Metagenome profiling is a very common approach of assessing microbial diversity from metagenomics data, which involves mapping unassembled reads or assembled contigs to reference databases in order to derive annotations for taxonomy, metagenomic pathways, genes, and other genomic content. Importantly, metagenome profiling requires the use of ≥1 reference database, which are all incomprehensive and are often biased toward pathogens and other well-studied microbes (Breitwieser, Lu & Salzberg, 2019; Loeffler et al., 2020). While microbial diversity can be directly assessed via kmer-based approaches that do not rely on reference databases (Benoit et al., 2016; Rowe et al., 2019), such methods provide no direct insight into the taxa or genomic content that differs across treatment groups or environmental gradients.

Most metagenome profiling software includes “standard” reference databases (Kim et al., 2016; Franzosa et al., 2018; Wood, Lu & Langmead, 2019; LaPierre et al., 2020). For instance, the popular HUMANnN pipeline includes multiple databases for assessing both taxonomy and function from read data (Franzosa et al., 2018). Similarly, Kraken2 includes a set of standard databases for taxonomic classification of specific clades (e.g., fungi or plants) or all taxa (Wood, Lu & Langmead, 2019). While such standard reference databases provide a crucial resource for metagenomic data analysis, they may be suboptimal for multiple reasons. First, databases will always be somewhat out-of-date, given the rapid increase in newly assembled genomic content. For instance, database biases and lack of representation can be reduced by first assembling genes and genomes from metagenomes and adding the new genomic content to the existing databases (Youngblut et al., 2020). Second, standard profiling databases may not be customized for specific research questions such as smaller databases focused on certain (well-curated) taxa or genomic content from specific environments (e.g., only marine or soil microbiomes). Third, the underlying taxonomy or genomic annotations included in the database may be suboptimal. For example, the Genome Taxonomy Database (GTDB) defines a taxonomy based on microbial genome phylogenies, which is highly standardized relative to the polyphasic taxonomy of the NCBI; however, the GTDB taxonomy is not available in standard profiling databases (Parks et al., 2020; De la Cuesta-Zuluaga, Ley & Youngblut, 2020).

The process of making custom reference databases is often complicated and requires substantial computational resources, which led us to create Struo for straight-forward custom metagenome profiling database generation (De la Cuesta-Zuluaga, Ley & Youngblut, 2020). We utilized our toolset to generate and publish custom metagenome profiling databases from the GTDB, which not only provides a standardized taxonomy but also increased the percent of gut metagenome reads mapped to database references relative to standard Kraken2 and HUMAnN2 databases.

While Struo greatly aids in custom database generation, it does require substantial computational resources, especially for the gene annotation that is necessary for creating custom HUMAnN databases. Specifically, Struo requires ∼2.4 CPU hours per genome and thus took >51,000 CPU hours when applied to ∼21,000 species-representative genomes in Release 89 of the GTDB. The number of representative genomes has increased to >31,900 in the subsequent GTDB release (an ∼50% increase), which would require >77,900 CPU hours (>9.1 years) for the Struo pipeline. Given that the number of genomes added to the GTDB and other databases is rapidly expanding (Fig. 1) (Almeida et al., 2020; Sayers et al., 2021), we sought to develop a pipeline that scales accordingly. Struo2 is a rewrite and expansion of Struo, which is substantially faster, can efficiently update existing databases, and includes generally applicable utility tools for database manipulation.

Figure 1 The number of microbial genomes and taxonomic lineages in the Genome Taxonomy Database (GTDB) is rapidly expanding.

(A) The number of total genomes and GTDB taxonomic lineages in each GTDB release. (B) The number of unique groups per taxonomic level in each GTDB release. Data was obtained from GTDB: https://gtdb.ecogenomic.org/downloads.

Materials & Methods

Struo2 database creation algorithm

A general outline of the Struo2 algorithm is shown in Fig. 2A. Struo2 can generate database files for four main database types: “Kraken2”, “Bracken”, “genes”, and “HUMAnN3” (Wood, Lu & Langmead, 2019; Lu et al., 2017; Franzosa et al., 2018). Struo2 uses snakemake and conda (Köster & Rahmann, 2012), and so there are no dependencies that must be installed prior to pipeline execution besides snakemake, conda, and pandas (for input table loading). Moreover, snakemake allows for efficient job execution and easy scaling on to high performance computing systems. We note that the Struo2 pipeline code is a substantial re-write and expansion of the original Struo pipeline (e.g., ∼1500 versus ∼7000 lines of code in Struo versus Struo2, respectively). Struo2 has extensive documentation, including tutorials on its usage.

Figure 2 Struo2 can build databases faster than Struo and can efficiently update the databases.

(A) A general outline of the Struo2 database creation algorithm. Cylinders are input or output files, squares are processes, and right-tilted rhomboids are intermediate files. The largest change from Struo is the utilization of mmseqs2 for clustering and annotation of genes. (B) Benchmarking the amount of CPU hours required for Struo and Struo2, depending on the number of input genomes. The equations show that Struo2 scales 3.5 fold faster than Struo1. (C) The number of genes annotated with a UniRef90 identifier, which shows that both Struo versions annotate approximately the same number of genes. (D) The percent of CPU hours saved via the Struo2 database updating algorithm versus de novo database generation. The original database was constructed from 1,000 genomes. For (B) and (D), the lines are linear regressions, and the grey regions represent 95% confidence intervals.

The user input for Struo2 database creation is a table that lists: (i) unique taxon names, (ii) assembly accession identifiers (if available), (iii) paths to (compressed) genome assembly fasta files, (iv) taxonomy identifiers (taxids) used for Kraken2 database construction, and (v) taxonomies at the genus and species levels (used for HUMAnN3). We provide 2 utility scripts to aid in construction of custom databases from genomes in the GTDB: GTDB_metadata_filter.R and genome_download.R. GTDB_metadata_filter.R can filter the publicly available GTDB archaeal and bacterial genome metadata files to a select subset of genomes (e.g., those with a lower CheckM-estimated contamination). genome_download.R can then download all of the user-selected GTDB genomes and add the path to the genome assembly fasta files to the GTDB metadata table. This updated metadata table can then be directly used as input to GTDB.

For construction of the custom Kraken2 database, contigs are renamed to “kraken:taxid— <taxid >— <seqid >”, as described in the Kraken2 manual (https://github.com/DerrickWood/kraken2/wiki/Manual). The renamed contigs are added to a new Kraken2 database via kraken-build, and then the database is constructed via the same command. By default, the GTDB taxonomy is used, which entails providing custom GTDB taxdump files created via the gtdb_to_taxdump.py utility tool (available at https://github.com/nick-youngblut/gtdb_to_taxdump). The “taxonomy” and “library” directories created by Kraken2 for temporary file storage are saved in order to expedite database updating with new genomes.

Custom Bracken database files are created for any number of read lengths that the user specifies (100 and 150 base pairs by default). The bracken-build.py script is used within the pipeline for constructing each Bracken database.

In order to construct a custom HUMAnN3 database, Struo2 first creates a precursor “genes” database, which consists of gene sequences from each genome and gene clusters generated via mmseqs linclust. To construct the “genes” database, genes are first called via prodigal (Hyatt et al., 2010), and then de-replicated at 97% sequence identity with vsearch (Rognes et al., 2016), which is similar to the standard HUMAnN database construction process (Franzosa et al., 2018). Non-redundant gene sequences from all genomes are combined, and the metadata of each gene sequence (e.g., genome of origin, contig of origin, and location on the contig) is also combined into one text file. The amino acid gene sequences are clustered via mmseqs linclust. By default, gene cluster representative sequences are annotated against UniRef90 (version 2019-01; the same as used by HUMAnN3) via mmseqs search with 2 search iterations and 3 sensitivity steps (min = 1, max = 6). Prior to annotation, the sequence queries are split into n batches and run in parallel for faster distributed searching with snakemake (n is user-defined). For each gene cluster, the UniRef90 annotations are propagated to each gene. UniRef90 annotations are mapped to UniRef50 identifiers via a mapping file created from the UniRef90.xml file available from UniProt (https://www.uniprot.org/uniref/). The unirefxml2clust50-90idx.py utility script is used to generate this mapping file (available at https://github.com/nick-youngblut/gtdb_to_taxdump). The mapping of UniRef90 to UniRef50 identifiers obviates the need to annotate genes separately against UniRef90 and UniRef50. We note that Struo requires separate rounds of annotation to each UniRef database instead of this UniRef90-to-UniRef50 mapping approach, which greatly increases the run time versus Struo2 when the goal is to obtain annotations for both UniRef90 and UniRef50. Note that the genes database includes both nucleotide and amino acid sequences for each gene.

The annotated gene sequences are renamed in the format “ <UniRefID >— <gene_length >—g__ <genus >;s__ <species >” for creation of the HUMAnN3 database. Note that the taxonomy information is provided by the user in the original input table. bowtie2-build and diamond makedb are used to generate a HUMAnN3-compatible bowtie2 and DIAMOND databases of all annotated gene nucleotide and amino acid sequences, respectively.

Struo2 database update algorithm

Struo2 can update existing Struo2-generated Kraken2, Bracken, genes, and HUMAnN3 databases. The databases can be updated with new genomes or individual gene sequences (e.g., created via metagenome assembly with PLASS (Steinegger, Mirdita & Söding, 2019)).

If the input is a set of new genomes, the input is essentially the same as for database creation, except the existing database files must also be provided. Database updating with individual gene sequences requires the gene sequences in amino acid format (and also nucleotide, if available) and metadata on each gene (i.e., the genus- and species-level taxonomy inferred via mmseqs taxonomy or other approaches).

Kraken2 custom databases are updated via adding more genomes to the existing library via kraken-build. New Bracken databases are created from the updated Kraken2 database.

Gene sequences, either originating from new genomes or new individual sequences, are added to the existing mmseqs gene cluster database via mmseqs clusterupdate. Newly formed clusters are annotated with mmseqs search, while existing annotations are used for existing clusters. The updated database of annotated genes are used for creating new HUMAnN3-compatible bowtie2 and DIAMOND databases.

We note that database updating does not require consistent genomic representation from each representative genome (e.g., the same taxonomic marker genes), given that neither Kraken2/Bracken nor HUMAnN3 require such consistency.

Benchmarking custom database construction and updating

We used genomes from the GTDB (Release 95) for all benchmarking.

Only genomes with ≥50% CheckM-estimated completeness, <5% CheckM-estimated contamination were included (Parks et al., 2015). To reduce biases towards species with large numbers of representative genomes, we selected one genome per species. The genome with the highest estimated completeness and lowest estimated contamination was selected for all candidates of each species. The final pool consisted of 30,989 genomes (Fig. S2).

We used the same genome subsets for benchmarking database creation with both Struo and Struo2. We benchmarked the combined time to generate Kraken2, Bracken, and HUMAnN databases, which included both UniRef50 and UniRef90 annotations for the HUMAnN databases. Struo was run with default parameters. Both pipelines were run on the same computational architecture, consisting of a high performance computing cluster comprising nodes running Ubuntu 18.04.5 with AMD Epyc CPUs and 0.5–2 terabytes of RAM. The CPU hours shown in Fig. 2B are the sum of all CPU hours for all snakemake jobs, as recorded via snakemake’s benchmarking feature.

We only benchmarked database updating for Struo2, given that Struo cannot update databases, and we also clearly show in Fig. 2B that database generation is much slower for Struo. We first used Struo2 to generate custom Kraken2, Bracken, and HUMAnN databases from 1000 genomes. These “n1000” databases were used for all database update benchmarking. The genomes used for database update benchmarking did not overlap with any genomes used to generate the n1000 databases, and they did not overlap with each other. We used subsets of 10, 100, 175, 250, 350, and 500 genomes. We used the linear regression models shown in Fig. 2B to estimate the CPU hours that would be required to generate each database from scratch rather than updating.

Benchmarking custom database accuracy as a function of genome assembly quality

We utilized two randomly chosen sets of 100 genomes from the GTDB r95 reference genome pool (described above). For each set, we simulated varying levels of misassemblies among each genome in the set, with the same number of misassemblies introduced into each genome per simulation set. Simulations were created via a custom python script: genome_mis-asmbl_sim.py (available in the Struo2 GitHub repository). Three types of misassemblies were simulated: breakpoints (splitting a contig into 2 pieces), rearrangements (relocating a genomic fragment within one contig to another location on the same contig or other contig in the assembly), and chimerisms (relocating a genomic fragment from one contig in a donor genome to a contig in a recipient genome). Breakpoint locations were selected from a uniform distribution. Fragment sizes of rearrangements and chimerisms were selected from a uniform distribution, with size a range of 1e3-1e4. We used CheckM to assess assembly quality of each genome in each synthetic genome dataset (Parks et al., 2015). Each resultant synthetic genome dataset was used to construct reference metagenome profiling databases via Struo2.

We utilized the Critical Assessment of Metagenome Interpretation (CAMI) “HMP Gut” metagenome dataset to assess how genome assembly quality affects Kraken2/Bracken taxonomic assignments (Sczyrba et al., 2017). The dataset was downloaded from the CAMI challenge website: https://data.cami-challenge.org/. The Illumina paired-end reads were subsampled to 1 million per sample. Beta diversity was calculated from the Bracken output via QIIME2 (Bokulich et al., 2018). Beta diversity distance matrices were compared via Mantel tests with 999 permutations and the Pearson correlation method.

Struo2 databases from GTDB releases 95 and 202

The genomes selected were as reported for the benchmarking of Struo and Struo2. The custom Kraken2, Bracken, genes, and HUMAnN3 databases, are available at: https://github.com/leylabmpi/Struo2/. We will publish new versions of each database as new releases of the GTDB are published.

Utility tools

We have generated a set of utility tools for aiding in the construction of input for Struo2 and generally facilitating the integration of the GTDB taxonomy into existing bioinformatics pipelines. Some of these tools are described elsewhere in the Supplement Methods. We note 2 utility tools that can have a broad applicability: gtdb_to_taxdump.py and ncbi-gtdb_map.py. The former can convert the GTDB taxonomy, as documented in the GTDB bacterial and archaeal metadata table, to NCBI-formatted taxdump files. These taxdump files can be used with any existing software that requires taxdump files, such as taxonkit (Shen & Xiong, 2019) or KrakenUniq (Breitwieser, Baker & Salzberg, 2018). ncbi-gtdb_map.py maps between NCBI and GTDB taxonomies, based on the taxonomy information provided in the GTDB archaeal and bacterial metadata files. This tool can be useful for converting GTDB-Tk classifications to NCBI taxonomies (Chaumeil et al., 2019), or converting existing NCBI taxonomies to GTDB taxonomies without requiring re-classification.

Code availability

The Struo2 pipeline code and all Jupyter Notebooks describing the analyses in this study are available on GitHub at https://github.com/leylabmpi/Struo2.

Results and Discussion

Struo2 generates Kraken2 and Bracken databases similarly to Struo (Lu et al., 2017; Wood, Lu & Langmead, 2019), but the algorithms diverge substantially for the time consuming step of gene annotation required for HUMAnN database construction. Struo2 performs gene annotation by clustering all gene sequences of all genomes using the rapid mmseqs2 linclust algorithm, and then each gene cluster representative is annotated via mmseq2 search (Fig. 2A; Supplemental Methods) (Steinegger & Söding, 2017; Steinegger & Söding, 2018). In contrast, Struo annotates all non-redundant genes of each genome with DIAMOND (Buchfink, Xie & Huson, 2015). Struo2 utilizes snakemake and conda, which allows for easy installation of all dependencies and simplified scaling to high performance computing systems (Köster & Rahmann, 2012).

Benchmarking on genome subsets from the GTDB showed that Struo2 requires ∼0.67 CPU hours per genome versus ∼2.4 for Struo (Fig. 2B), which is a >3.5 fold decrease. Notably, Struo2 annotates slightly more genes than Struo, possibly due to the sensitivity of the mmseqs search iterative search algorithm (Fig. 2C). The use of mmseqs2 allows for efficient database updating of new genomes and/or individual gene sequences via mmseqs clusterupdate (Fig. S1); we show that this approach saves 15–19% of the CPU hours relative to generating a database from scratch, with a linear trend towards increased efficiency as the number of genomes are added (Fig. 2D).

We assessed how Struo2-generated databases can be affected by the assembly quality of the reference genomes used. For this assessment, we generated synthetic reference genome datasets, in which 3 types of misassemblies were introduced into each genome: breakpoints, relocations, and chimerisms (see Methods). As we increased the number of per-genome misassemblies, CheckM-estimated completeness and contamination values substantially declined and increased, respectively (Fig. 3A). The percent of annotated genes, relative to the ground truth reference genomes, declined linearly and steeply as median assembly completeness declined (linear regression, n = 10, R2 = 0.97) and contamination increased (linear regression, n = 10, R2 = 0.9; Fig. 3B). Even at a median completeness of 95% and a median contamination of 1.5%, the percent of genes annotated was reduced by 25% relative to the ground truth, indicating that assembly quality has a substantial impact on gene database quality. The percent of correctly annotated genes (i.e., the correct UniRef90 ID versus the ground truth) was also linearly associated with assembly quality (linear regression, n = 10, completeness: R2 = 0.87, contamination: R2 = 0.7), but the decline in annotation quality was more severe: only ∼60% of genes were correctly annotated when median completeness dropped to ∼92% (Fig. 3C). In contrast, Kraken2/Bracken database quality was not substantially affected by assembly quality, in regards to beta diversity relative to the ground truth (Fig. 3D). Beta diversity divergence increased only slightly as misassemblies increased (Mantel, permutations = 999, ρ > 0.98 for all tests). The robustness of the Kraken2/Bracken databases is likely due to the use of kmers derived from entire genomes instead of using annotated genes, as used by HUMAnN3. In summary, we recommend using high quality assemblies for custom database construction, especially for the creation of the HUMAnN3 database.

Figure 3 Struo2-generated gene database quality is substantially affected by reference genome assembly quality.

Two reference genome datasets of 100 randomly selected genomes each (“n100_r1” and “n100_r2”) were used for simulating misassemblies among all genomes in order to assess how genome assembly quality affects Struo2-generated database quality. “Ground truth” is the unaltered reference genomes, while the “bN-rN-cN” labels denote synthetic datasets with specific numbers of added misassemblies per genome (see Methods). (A) CheckM-estimated assembly quality for each genome. (B) The percent of genes annotated in the Struo2 database versus the ground truth. (C) The percent of genes annotated correctly (i.e., correct UniRef90 ID) versus the ground truth. (D) Change in Bray-Curtis distances between the ground truth and synthetic datasets (measured via Mantel tests), with beta diversity calculated from Bracken taxonomic assignments. The CAMI2 “HMP-gut” dataset of 10 metagenomes was used for benchmarking.

We used Struo2 to create publicly available Kraken2, Bracken, and HUMAnN3 custom databases from releases 95 and 202 of the GTDB (see Supplemental Methods). We note that the reference genomes selected from releases 95 and 202 had a median CheckM-estimated completeness of 98.5 and 97.3%, respectively. The median estimated contamination was 0.71 and 0.79%, respectively. Thus, our custom GTDB databases should be of high quality. We will continue to publish these custom databases as new GTDB versions are released. The databases are available at https://github.com/leylabmpi/Struo2. We also created a set of utility tools for (i) generating NCBI-formatted taxdump files from the GTDB taxonomy, (ii) mapping between the NCBI and GTDB taxonomies, and (iii) generating DIAMOND databases for any set of GTDB reference genomes. The taxdump files are utilized by Struo2, but these tools can be used more generally to integrate the GTDB taxonomy into existing pipelines designed for the NCBI taxonomy (available at https://github.com/nick-youngblut/gtdb_to_taxdump).

In summary, Struo2 provides a substantial improvement over the state-of-the-art, which is needed in order to scale custom database generation with the ever-increasing amount of available genomic data. The efficient database updating feature of Struo2 enables the following “assembly enhanced” taxonomic profiling workflow. First, the user assembles MAGs and/or genes from the metagenomes novel to the user’s own study. Second, the user updates our pre-generated GTDB-based profiling databases with the newly generated genomic content. Third, the user taxonomically profiles the metagenome reads from the user’s study against the newly customized databases in order to generate microbial diversity assessments that are less biased and more representative of the microbiome diversity in the microbiomes included in the study. This “assembly-enhanced” profiling method could greatly improve database representation for less-studied environments such as the gut microbiome of non-mammalian vertebrates or under-represented human populations (Porras & Brito, 2019; Youngblut et al., 2020).

Supplemental Information

Supplemental Information 1 An overview of the Struo2 algorithm for database updating

Cylinders are input or output files, squares are processes, and right-tilted rhomboids are intermediate files. Existing Kraken2, Bracken, genes, and HUMAnN3 databases can be updated with new genomes, while only existing genes and HUMAnN3 databases can be updated with new individual gene sequences.

Click here for additional data file.

Supplemental Information 2 The number of GTDB genomes per phylum used for Struo2 generation of the custom Kraken2, Bracken, genes, and HUMAnN3 databases available at http://ftp.tue.mpg.de/ebio/projects/struo2/

See the Materials & Methods for information on how genomes were selected. The phylum names shown are based on the GTDB taxonomy.

Click here for additional data file.

We thank Albane Ruaud, Liam Fitzstevens, Jacobo de la Cuesta-Zuluaga, and Jillian Waters for providing helpful comments on an earlier version of this manuscript.

Additional Information and Declarations

Competing Interests

Author Contributions

Data Availability

The authors declare there are no competing interests.

Nicholas Youngblut conceived and designed the experiments, performed the experiments, analyzed the data, prepared figures and/or tables, authored or reviewed drafts of the paper, and approved the final draft.

Ruth Ley conceived and designed the experiments, authored or reviewed drafts of the paper, and approved the final draft.

The following information was supplied regarding data availability:

Struo2 code: https://github.com/leylabmpi/Struo2.

Pre-built metagenome profiling databases: https://github.com/leylabmpi/Struo2.

Utility tools for taxonomy and database processing: https://github.com/nick-youngblut/gtdb_to_taxdump.

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
