# Peer review of "Struo2: efficient metagenome profiling database construction for ever-expanding microbial genome datasets"

_PeerJ, doi:10.7717/peerj.12198_

## Round 0.1 · accepted · Accept

Dear Dr. Youngblut,

Both reviewers found the manuscript to be of high quality and to have adequately addressed the concerns of previous reviewers. I agree with this assessment and the importance of this work given the rapidly increasing number of genome assemblies. I look forward to seeing the published manuscript.

Regards,
Donovan Parks

Reviewer 1 ·

Basic reporting

The manuscript is well structured, straightforward and easy to read. Enough background is provided to contextualize the work. Figures are clear and references are up-to-date.

Experimental design

Methods are well-described both in the manuscript and in the associated GitHub repository. The authors have done a very thorough effort of describing each individual step to ensure reproducibility.

Validity of the findings

The conclusions regarding the speed of the tool and the effect of assembly quality on taxonomic classification are well supported. As detailed below, I would be interested in understanding why database “b100-r20-c10” in their simulation showed reduced performance in the Kraken2/Bracken test (Fig. 3D) compared to the other sets (even though there were others of lower assembly quality).

Additional comments

Youngblut and Ley describe an updated version of Struo, a software tool for constructing custom databases for commonly used metagenomic profilers. Their newer version is more than 3 times faster and has additional flexibility to update existing databases. This works represents a very useful contribution to the metagenomics field, as there is an increasing need to develop bioinformatics tools that can scale with the hundreds of thousands of microbial genomes that are now available in the public domain. In addition, given that the Genome Taxonomy Database is becoming the gold-standard approach for genome classification, it is especially useful to be able to generate custom databases using this updated taxonomic framework.

I did not review the initial version of the manuscript, but I believe the authors have appropriately addressed the reviewers’ comments and suggestions. I commend the authors for the extensive documentation provided in the GitHub repository, as it will be particularly useful for others to be able to easily run and test their tool. I only have two suggestions/comments:

1) Would be important to discuss in the text any useful pre-processing that should be done on the genomes prior to database building. The authors evaluated the effect of assembly quality on classification accuracy, but do all the tools cope with a highly redundant genome database, or should genomes be dereplicated (e.g. at 95% identity) for increased performance? Dereplication at the protein level appears to already be embedded within the HUMAnN3 database construction. What effect would different genome clustering thresholds have on taxonomic profiling?

2) Their Kraken2/Bracken benchmarking with the simulated databases is quite useful. However, I am wondering why database b100-r20-c10 showed the worst performance (Fig. 3D), even though it was of better assembly quality than some of the other sets? Could the authors elaborate on which specific genomes affected the results and why?

Reviewer 2 ·

Basic reporting

no comment

Experimental design

no comment

Validity of the findings

no comment

Additional comments

I have carefully reviewed the manuscript from Youngblut & Ley entitled "Struo2: efficient metagenome profiling database construction for ever-expanding microbial genome datasets", which came to me already after its exhaustive response to reviewers, which I also assessed. I feel that authors answered satisfactorily and exhaustively all previous reviewer's comments, and the questions I personally also had (mainly about the inclusion of Metaphlan in the Struo2 pipeline) were also raised by some reviewers and answered satisfactorily by authors. This manuscript is strong and provides a much needed tool aiding the analysis of metagenomes, very importantly so for the easy inclusion of GTDB definitions in commonly accepted metagenome analysis pipelines.